Treemendous: an R package for integrating taxonomic information across backbones

Specker Felix 1 2
Paz Andrea 1
Crowther Thomas W. 1
Maynard Daniel S. dan.s.maynard@gmail.com 1 3
1 Institute of Integrative Biology, ETH Zürich , Zürich , Switzerland
2 Department of Biosystems Science and Engineering, ETH Zürich , Zürich , Switzerland
3 Department of Genetics, Evolution and Environment, University College London , London , United Kingdom
Fischer Daniel
Electronic publication date: 2024 Feb 28
Publication date: 2024
Volume: 12
Electronic Location ID: e16896
Received 2023 May 2; Accepted 2024 Jan 16
Copyright: ©2024 Specker et al.
Copyright year: 2024
Copyright holder: Specker et al.
License: This is an open access article distributed under the terms of the Creative Commons Attribution License, which permits unrestricted use, distribution, reproduction and adaptation in any medium and for any purpose provided that it is properly attributed. For attribution, the original author(s), title, publication source (PeerJ) and either DOI or URL of the article must be cited.
License URL: https://creativecommons.org/licenses/by/4.0/

Keywords: Biodiversity research, Forest inventory, Nomenclature, R language, Taxonomic databases

Funding: The Swiss National Science Foundation #PZ00P3_193612 DOB Ecology and the Bernina Initiative Daniel Maynard and Andrea Paz were supported by an Ambizione grant from the Swiss National Science Foundation to Daniel Maynard (#PZ00P3_193612). Thomas Crowther was supported by grants from DOB Ecology and the Bernina Initiative. The funders had no role in study design, data collection and analysis, decision to publish, or preparation of the manuscript.

==============================
Standardizing and translating species names from different databases is key to the successful integration of data sources in biodiversity research. There are numerous taxonomic name-resolution applications that implement increasingly powerful name-cleaning and matching approaches, allowing the user to resolve species relative to multiple backbones simultaneously. Yet there remains no principled approach for combining information across these underlying taxonomic backbones, complicating efforts to combine and merge species lists with inconsistent and conflicting taxonomic information. Here, we present Treemendous, an open-source software package for the R programming environment that integrates taxonomic relationships across four publicly available backbones to improve the name resolution of tree species. By mapping relationships across the backbones, this package can be used to resolve datasets with conflicting and inconsistent taxonomic origins, while ensuring the resulting species are accepted and consistent with a single reference backbone. The user can chain together different functionalities ranging from simple matching to a single backbone, to graph-based iterative matching using synonym-accepted relations across all backbones in the database. In addition, the package allows users to ‘translate’ one tree species list into another, streamlining the assimilation of new data into preexisting datasets or models. The package provides a flexible workflow depending on the use case, and can either be used as a stand-alone name-resolution package or in conjunction with existing packages as a final step in the name-resolution pipeline. The Treemendous package is fast and easy to use, allowing users to quickly merge different data sources by standardizing their species names according to the regularly updated database. By combining taxonomic information across multiple backbones, the package increases matching rates and minimizes data loss, allowing for more efficient translation of tree species datasets to aid research into forest biodiversity and tree ecology.

Background

Large-scale biodiversity research often requires combining different data types, such as occurrence, genetic, and trait information, from across a variety of public and private sources (Thomas, 2009). Disagreeing taxonomic backbones or different spelling variants of species can pose major difficulties and force researchers to spend many hours on manual annotations. In order to streamline biodiversity research, maintaining a consistent taxonomy of species names and offering tools to overcome the associated challenges is crucial (Grenié et al., 2021). A plethora of tools have been published in recent years, including online access to several databases, list matching, and data wrangling for a diversity of taxa and data types (Grenié et al., 2021). However, when homogenizing the different datasets, loss of data is still a problem if using exact matching or selecting a single backbone out of the various recognized ones. Using combinations of different tools to overcome this challenge can be time-consuming, difficult to reproduce, and computationally intensive.

Taxonomic name resolution can be divided into three general parts: (1) name parsing, where input names are preprocessed to fix formatting issues and obtain a standardized input format (e.g., a Latin binomial or trinomial); (2) name matching, where the parsed names are cross-referenced with a taxonomic list, often with the aid of so-called “fuzzy matching” to account for spelling mistake and variations; and (3) name resolution, where the matched names are resolved to an accepted species, based on the relationships present in a focal backbone. Over the last decade, numerous high-quality name resolution packages have become available, with each one typically focusing on a different aspect of this workflow. For example, the Global Name Parser (Mozzherin, Myltsev & Patterson, 2017) focuses on Step 1, parsing scientific names into the different semantic elements, including annotations, taxonomic ranks, authorship, and so on, along with the associated metadata. Alternatively, Taxamatch (Rees, 2014) focuses on Step 2, providing a set of fast and efficient name-matching algorithms to handle misspellings while hierarchically matching genus, species, authorship, and rank to a focal backbone. Finally, applications such as taxonomic name resolution service (TNRS) have extensive functionality for all three steps, providing multiple different backbones for name resolution and identification of accepted species, and even identifying the optimal match across these different backbones (Boyle et al., 2013).

What remains an open challenge, however, is the ability to combine and resolve conflicting information across backbones or species lists. When integrating new data into an analytical workflow or database, the user is frequently confronted with the need to merge new names into an existing taxonomic list, which is often inconsistent with any given backbone. In many cases, these lists include names with fundamentally different taxonomic origins, such as historical datasets with outdated or idiosyncratic taxonomic lists that are often specific to local regions. The assembly of massive, highly curated backbones has helped to overcome this challenge by comprehensively identifying and cataloging homotypic synonyms, infraspecific names, name variants, and misspellings, and linking them into an accepted Latin binomial and author. While such databases have proved indispensable in biodiversity research, there now exist numerous competing backbones with varying levels of accepted species and synonyms. To illustrate, the Kew Gardens’ “World Checklist of Vascular Plants” contains upwards of 343,000 plant species and 1,020,000 synonyms (Govaerts et al., 2021); compared to “World Flora Online” with 382,000 species and 1,420,000 synonyms (Borsch et al., 2020). While similar, these backbones disagree on over 40,000 species (>10%) and contain a difference of at least 400,000 synonyms (>25%). Indeed, recent comparisons across taxonomic checklists suggest agreement in approximately 60% of taxonomic names (Schellenberger Costa et al., 2023), which, for any taxonomic list of considerable size, leads to substantial mismatches and omissions regardless of the target backbone selected.

The compilation of these high-quality curated backbones has thus created a new problem for the end user: the current name-resolution workflow requires one to identify the “best” backbone for their specific problem, despite the fact that their data are unlikely to be fully compatible with any given backbone. To address this, applications such as U.Taxonstand (Zhang & Qian, 2023), TNRS (Boyle et al., 2013) or the earlier taxize (Chamberlain & Szöcs, 2013) can be used to query multiple different backbones, either iteratively searching for additional matches for names not found in the first backbone, or by providing an internally suggested best match based on expert knowledge from the application designers. This approach certainly increases the success rate, but it does not consider relationships among synonyms across the backbones when resolving names. To illustrate the challenge, suppose that a user has a species list containing species A and B that they wish to harmonize with Backbone 1. The user first queries this backbone, and finds that A is an accepted species, but B has no match. The user then queries Backbone 2 for species B, and finds it is a synonym for the accepted species C. The user is thus left with two accepted species, A and C. But if species C is present in Backbone 1 as a synonym of A, then this approach is inconsistent with the first target backbone by falsely giving two unique species. This problem not only hinders the merging of species lists, but it complicates future integration of new data since the resulting list is not easily reproducible and depends on idiosyncratic user-defined matching routines. In order to overcome this limitation, taxonomic name resolution services need to provide a principled approach to mapping the connections among names across backbones, enabling one to incorporate all available information when translating a species list into a target backbone.

Here, we introduce the package Treemendous, which presents a novel fourth step in the name-resolution pipeline that incorporates relationships across backbones when resolving species names. It does so by implementing a graphical approach to link all synonyms and species across multiple backbones to find the shortest path between taxa in an input dataset and an accepted species in a target backbone. In addition to leveraging information across backbones to improve matching success, this approach also allows one to directly “translate” one species list into another reference list, even if the reference list is inconsistent with any given backbone. The package also provides basic name-cleaning and name-matching functionality, allowing it to serve as a standalone package, but more ideally it is intended to be used in tandem with other name-parsing and name-cleaning packages as a downstream fourth step in the name-resolution workflow. Our focus here is on trees rather than all plants primarily for data quality and data size limitations, but the functionality of the program can be extended to any taxonomic group with user-supplied backbones. The package accordingly comes with the database Treemendous.Trees, consisting of tree species compiled from four publicly available databases. We first provide a description of the package and associated database Treemendous.Trees and how it was created, followed by a description of the core functions of the package along with a worked-out example of the package functionality.

Package description

The Treemendous package is developed using the R statistical software (R Core Team, 2022) and is relying heavily on the collection of well-maintained packages available through tidyverse (Wickham et al., 2019). Figure 1 shows the major components of the Treemendous package. The functionality of the package is divided into two main steps: First, the names are matched against the database Treemendous.Trees using either the function matching() or the function sequential_matching(). The latter is used when the input names should be matched sequentially against Treemendous.Trees, i.e., according to a user-specified ordering of backbones. Optionally, the user can try to increase the proportion of matched species to a single backbone using enforce_matching(). Afterwards, the matched names can then be resolved using resolve_synonyms(), replacing synonyms with their respective accepted species names. Additionally, translate_trees() allows users to translate an input species list into a custom target database, making use of the synonym-accepted relations in Treemendous.Trees. Finally, an overview of the process information can be obtained with the function summarize_output(). In the following sections, the databases and main functions are described in greater detail, along with several examples demonstrating common usage.

The core novel functionality of this package is found in the enforce_matching(), resolve_synonyms(), and translate_trees() functions, with the additional upstream functions providing basic usage for Steps 1–3 of a name resolution workflow. These downstream functions can also be applied to previously parsed and cleaned data (though note that one of the matching functions must first be applied before resolving or translating), allowing the user to integrate the novel aspects of this package into previous name resolution pipelines.

Access & Installation

Treemendous is an open-source package hosted on GitHub and is freely available at https://github.com/speckerf/treemendous. The package can be installed in R using the devtools package (Wickham et al., 2021a) by calling

devtools::install_github(”speckerf/treemendous”).

Alternatively, this package is available as a stand-alone docker image, containing all packages and dependencies; see https://github.com/speckerf/treemendous for installation details.

Figure 1 Overview of the functionality of the Treemendous package.

First, the species names are matched using either matching() or sequential_matching(). Optionally, enforce_matching() can be called afterwards. Synonyms can be resolved using resolve_synonyms(). The function summarize_output() summarizes the process information.

Dependencies

All R package dependencies are installed (if missing) along with the base installation of Treemendous. Every function in Treemendous requires the input to be a tibble (Müller & Wickham, 2022). String manipulations are performed using stringr (Wickham, 2019b) and stringi (Gagolewski, 2022). The package purrr (Henry & Wickham, 2020) is used for functional programming. The package readr (Wickham, Hester & Bryan, 2022) is used to import the data for the Treemendous.Trees database. Throughout the whole package, dplyr (Wickham et al., 2022) and tidyr (Wickham & Girlich, 2022) are extensively used for working with tibbles. Progress bars are implemented in progress (Csárdi & FitzJohn, 2019) and speed-ups are achieved using memoise (Wickham et al., 2021b), which can save outputs of utility functions to memory and reload them upon the second function call with equivalent arguments. The pipe operator provided via magrittr (Bache & Wickham, 2022) is used to increase the readability of the code, and assertthat (Wickham, 2019a) is used to increase code safety.

The Treemendous.Trees database

The treemendous package uses an internal database called Treemendous.Trees for its functionality. Treemendous.Trees contains 401,482 different tree species names assembled from four different publicly available databases. This includes synonyms and spelling variants, which are present in the underlying databases. Further, because tree identification is implemented at the genus level, the database contains a significant number of species that may not meet the requirements of being a tree by some definitions (e.g., are a small woody shrub or woody vine), but which are included in the backbone because they occur in a genus containing a tree. This inclusive approach allows this package to be flexibly applied to scenarios with differing definitions for trees vs. other woody plants.

To construct Treemendous.Trees, we used the GlobalTreeSearch database published by the Botanical Gardens Convention International(BGCI) (Beech et al., 2017) to compile a list of taxonomical genera containing at least one tree species (n = 4,189). All genera were used to extract potential tree species from the publicly available databases of World Flora Online(WFO) (Borsch et al., 2020), the World Checklist of Vascular Plants(WCVP) (Govaerts et al., 2021) and the Global Biodiversity Information Facility(GBIF) (GBIF Secretariat, 2021). After filtering these databases by the list of tree genera, we kept all entries with rank ‘Species’ and with the corresponding taxonomical status being ‘Accepted’, ‘Synonym’, ‘Homotypic Synonym’ or Heterotypic Synonym’. Synonyms in these databases always contain information about which species is their corresponding accepted name.

We then ensured the database is self-contained, meaning that for every synonym the corresponding accepted species are included in Treemendous.Trees. First, for all accepted species names, synonyms with a genus different from the list of tree genera were added. Second, for all synonym species names, the corresponding accepted specie(s) were included as well, even if their genus was not in the BGCI list of tree genera. Table 1 shows the taxonomical status of the species from WFO, WCVP and GBIF, which have been included in Treemendous.Trees.

Table 1 Crosstabulation of the taxonomic status of the species of Treemendous.Trees from the three backbones of World Flora Online (WFO), The World Checklist of Vascular Plants (WCVP) and the Global Biodiversity Information Facility (GBIF).

In each database, a species name can be present and considered as accepted or not accepted or missing. A total of 124,044 species are accepted in all three databases. Note: The total number of accepted species in each database is higher than the estimated global number of tree species because some genera might contain both tree and non-tree species. 140 species are absent from these three backbones, but present in BGCI (Botanic Gardens Conservation International), which considers all species as being accepted.

		GBIF	Accepted	Not accepted	(Missing)	
WFO	WCVP					
Accepted	Accepted		101,348	1,710	25,934	
	Not Accepted		2,275	837	3,433	
	(Missing)		1,830	382	2,635	
Not Accepted	Accepted		739	288	1,020	
	Not Accepted		5,905	39,332	108,555	
	(Missing)		1,228	5,433	17,406	
(Missing)	Accepted		6,249	167	3,371	
	Not Accepted		1,182	2,914	14,174	
	(Missing)		18,040	37,252	0	

Currently, the database and corresponding backbones are periodically updated and assigned a corresponding GitHub release number, allowing the user to track versions and facilitate reproducibility.

Datasets

BGCI.

The Botanic Gardens Conservation International (BGCI) network is formed by botanic gardens from more than 100 countries. In 2017, BGCI published the GlobalTreeSearch database (Beech et al., 2017) with represents the most widely adopted and curated global list of tree species. The database uses the tree definition of the IUCN’s Global Tree Specialist Group: “A woody plant with usually a single stem growing to a height of at least two metres, or if multi-stemmed, then at least one vertical stem five centimetres in diameter at breast height.” The BGCI dataset contains 4,189 distinct genera and 57,921 tree species names. All these names are considered to be accepted species names and treated as such by the Treemendous package. BGCI regularly updates their list, with the version 1.7 (https://tools.bgci.org/global_tree_search.php) (April, 2023) in Treemendous version 1.1.1.

WFO.

The World Flora Online (WFO) is a portal of scientifically verified biodiversity data on bryophytes, pteridophytes, gymnosperms and angiosperms (Borsch et al., 2020). The WFO published an actively curated Taxonomic Backbone, which is a synonymized checklist of more than a million plant species. Every synonym in this checklist comes along with information about its accepted species name. In total, we extracted 346,427 (potential) tree species as previously described, of which 144,318 are considered to be accepted species. Obviously, this also includes many species which are not strictly considered trees, but rather shrubs and other plants. WFO regularly updates their species list, with the version v.2023.06 (http://www.worldfloraonline.org/downloadData) (June, 2023) in Treemendous version 1.1.1.

WCVP.

The World Checklist of Vascular Plants (WCVP) is a global consensus view of all known vascular plant species (flowering plants, conifers, ferns, clubmosses and firmosses) (Govaerts et al., 2021). It is managed by the Royal Botanic Gardens, KEW, and contains around a million taxonomical names at the species level, with information about the synonymy of names. In total, we extracted 344,165 (potential) tree species as previously described, of which 140,368 are considered to be accepted species.

WCVP regularly updates their species list, with the version v9 (http://sftp.kew.org/pub/data-repositories/WCVP/Archive/) (June, 2022) in Treemendous version 1.1.1.

GBIF.

The Global Biodiversity Information Facility (GBIF) is an international network and was created in 1999 after the OECD had stated the need for a central and free provider of biodiversity data (Muller, 2004). The GBIF Backbone Taxonomy (GBIF Secretariat, 2021) unifies various data sources and provides a single backbone for all life on earth, containing more than six million records. The backbone is created by using the Catalogue of Life (Bisby et al., 2012) as a starting point and trying to integrate more than 500 different data sources (GBIF Secretariat, 2021). In total, we extracted 371,483 (potential) tree species as previously described, of which 161,347 are considered to be accepted species. GBIF regularly updates their backbone, with the version from December 2022 (https://hosted-datasets.gbif.org/datasets/backbone/) in Treemendous version 1.1.1.

In order to display the current versions of all the backbones, please type ?Treemendous.Trees.

Functions for standardizing species names

When using the Tremendous package, the process of name resolution is divided into multiple steps. First, the species names are matched using either matching() or sequential_matching(). Optionally, enforce_matching() can be called afterwards. These functions provide matches in the target database/s regardless of the status of the match as an accepted name or synonym. If the user wants only accepted names to be returned, then synonyms can be resolved using resolve_synonyms() after using the matching functions. Please keep in mind that a species name can have multiple matches if there are authorship ambiguities or infraspecific matches (e.g., it matches to a species and also a variety).

The function highlight_flags() can be used to get information on flagged records. The function summarize_output() summarizes the process information.

matching(): The function requires the user to provide the species names as a tibble (https://tibble.tidyverse.org/), containing the genus and the specific epithet as two different columns; column names should be Genus and Species. Optionally, the user can specify the backbone, which can be any subset of c(”BGCI”, ”WFO”, ”WCVP”, ”GBIF”) and will filter the Treemendous.Trees database by the selected backbones. If no backbones are specified, the whole Treemendous.Trees database consisting of species from all four backbones is used for matching.

An overview of the functions called by matching() is shown in Fig. 2. First, direct_match() is called, and if the exact same name (genus and specific epithet) is present in the database then a match is produced. If there was no direct match, genus_match() checks, if the genus exists in the database. If the genus was not present, fuzzy_match_genus() is called, and this function, tries to inexactly match genus names using the package fuzzyjoin (Robinson, 2020) based on an optimal string alignment distance of one, as implemented in stringdist (van der Loo, 2014). In addition to insertions, deletions and substitutions, the metric also considers transpositions (e.g., Quercus ↔ Quecrus) as operations of distance one. If more than one genus matches, they are sorted alphabetically and the first match is picked, but the user is informed and encouraged to curate the ambiguous entries by hand. The maximal genus edit distance is set to one by design, because typos in genus names can be considered much rarer compared to the specific epithet and because genus names are usually quite short.

Figure 2 Overview of the matching() function: The process is split into six functions, which match the names against the full database Treemendous.

Trees or based on specific backbones.

After the genus name has been matched, three functions are called within the matched genus. First, direct_match_species_within_genus() checks if the specific epithet is present in the matched genus. If not, suffix_match_species_within_genus() tries to capture gender-specific endings or other common suffixes. More specifically, the following suffixes are substituted c(”a”, ”i”, ”is”, ”um”, ”us”, ”ae”). Next, the remaining unmatched species names are fuzzy matched with a maximal optimal string alignment distance of two.

The function matching() returns a tibble, with the new columns Matched.Genus and Matched.Species containing the matched names, or NA if there was no match. Further, a logical column is added for every function called to allow the user to inspect which functions were used for every name during the process. When a logical column shows NA, this function was not called for the given name because it was already matched with a preceding function. Please note that, in order to obtain only accepted names, the function resolve_synonyms() must be called (Figs. 3A, 3B).

Figure 3 Example of species name resolution using three different backbones in A–C, and a detailed example of the internal process behind the enforce_matching() function in D.

The GBIF backbone is not included in the figure as it behaves exactly as the WFO backbone in this case. Input species names are shown in blue, while output names are shown in green and intermediate names in grey. The four input species are Parinari racemosa, Atuna racemosa, Atuna excelsa and Maranthes corymbosa. (A) For the WFO backbone, all species have direct matches. After matching, resolve_synonyms() is used to obtain accepted species names according to the target backbone, WFO. (B) For the WCVP backbone all species have direct matches. After matching, resolve_synonyms() is used to obtain accepted species names according to the target backbone, WCVP. (C) For the BGCI backbone, the three species do not have direct matches and thus enforce_matching() is used. Because the resulting species are all accepted in BGCI, the subsequent use of resolve_synonyms() is not necessary. (D) When using enforce_matching(), input names not present in the target backbone (BGCI in this case, see (C)) can be matched using the information of synonym-accepted relations from the other backbones, at a depth of 1 in the case of Parinari racemosa, 2 in the case of Atuna excelsa, and 3 in the case of Atuna racemosa. According to WCVP, Parinari racemosa is considered a synonym of Maranthes corymbosa (see B), which is in our target database, BGCI. The species Atuna racemosa can be matched to our target database, BGCI, via the intermediate species names Atuna excelsa and Parinari racemosa, because WFO connects these species names (see A). The species name Atuna excelsa can be matched to BGCI via the intermediate species name Parinari racemosa, because WFO connects these species names (see A). Both these species can then be matched to Maranthes corymbosa, a species present in our target backbone, BGCI, through their connection in WCVP.

sequential_matching(): If the user wants to enforce an ordering upon the individual backbones, the function sequential_matching() can be used and the ordering is specified with the argument sequential_backbones. This function is a wrapper around matching() and calls it sequentially for every backbone in sequential_backbones. To ensure that the correct information on the functions used for all unmatched species, matching() is called again with all backbones together. Otherwise, the information would still correspond to the last backbone in sequential_backbone. Please note that in order to obtain only accepted names the function resolve_synonyms() must be called.

enforce_matching(): This function provides the first novel extension of this package. After having called either matching() or sequential_matching(), the user can optionally call enforce_matching(), trying to increase the proportion of matched species according to a single target backbone (see Fig. 3C). The function makes use of all the relations between synonyms and accepted species present in the backbones WFO, WCVP and GBIF. Using the package igraph (Csardi & Nepusz, 2006), an undirected graph g is created, with vertices representing species names, and edges indicating that two species names are considered synonymous according to a backbone. Additionally, two species names that can be matched via fuzzy-matching (maximum string-dist of one) are also connected with an edge.

Next, the algorithm iteratively tries to find a path from the input species to a species in the target backbone (see Fig. 3D). For multiple matches, the algorithm always selects the first match, i.e., the target vertex with lower ID_matched in Treemendous.Trees to ensure reproducibility. By default, the function allows a maximum depth of three steps to search for a match in the target backbone (see Fig. 3D), with the output field enforced_matching_dist denoting the depth of the match for each species (1, 2, or 3). Filtering by this column allows the user to be more restrictive (depth =1), at the cost of incorrectly missing some matches, or be increasingly permissive with the matches (depth =2 or 3), at the cost of potentially lumping species together. Depending on the application, these different scenarios may be more or less preferable and can be selected on a case-by-case basis. Note that as soon as a first match is found, the algorithm does not continue to look for matches at greater depths. Please note that in order to obtain only accepted names the function resolve_synonyms() must be called.

Although enforce_matching() represents a very powerful tool, the user is encouraged to manually check that these matches are also reasonable for their individual use case, and to explore the output of highlight_flags() (described below) to investigate input taxa with questionable or conflicting results.

resolve_synonyms(): This function works in tandem with the three matching functions to resolve synonyms to an accepted Latin binomial in the desired focal backbone, while leveraging synonym relationships across all backbones. As many of the species names in Treemendous.Trees are not considered to be accepted, but synonyms, the user might want to resolve these names according to a certain backbone. Three backbones (WFO, WCVP, GBIF) provide information about the accepted species name, while the species of BGCI are considered to only represent accepted names. The function resolve_synonyms() requires that the names were matched beforehand, using either matching(), enforce_matching(), or sequential_matching(), and it takes the result of these functions as an input (see Figs. 3A, 3B).

Because there is no consensus about which species is accepted among the databases, the user has to specify an order in these backbones. By default, the order is c(‘BGCI’, ‘WFO’, ‘WCVP’, ‘GBIF’), which can be modified by providing a different order via the argument backbones. By design, every species considered a synonym according to WFO, WCVP or GBIF, has the corresponding accepted species as part of the database.

translate_trees(): This function provides the second major contribution of this package, allowing the user to translate the species names of an input dataset into an existing custom target backbone, which need not be consistent with any underlying backbone. Essentially, the function is a wrapper around matching() and enforce_matching(), which at runtime merges the built-in Treemendous.Trees database with the provided custom backbone. This allows the function enforce_matching() to use the information of synonym-accepted relations in Treemendous.Trees, even when a custom target backbone is provided. In more detail, we first match the input names to the target names, including suffix and fuzzy matching. For all unmatched species, we match these to the species names present in the graph g representing synonym-accepted relations of Treemendous.Trees, thereby getting the entry vertices in g. Further, we find all vertices in the graph g, which can be matched to the new target database—essentially representing the target vertices in the graph g. Then, the function enforce.matching tries to find a path between the entry and the target vertices with a maximum depth of three (by default). By taking a graph-based approach, this function allows users to match a focal list of species to a target list of species, even if that target list is not fully consistent with any of the individual backbones (e.g., it contains species names that are not present in any one backbone).

highlight_flags(): The Treemendous package currently only uses Latin binomials as inputs and outputs, but the underlying database was constructed using taxonomic authority and infraspecific information to identify the types of linkages. The highlight_flags() function uses this metadata to flag resolved matches that have conflicting or potentially dubious results. In particular, the backbones WFO, WCVP or GBIF often contain multiple entries for the same Latin binomial, which, in the absence of authority information, would resolve to different Latin binomials using resolve_synonyms(). In other cases the input is both an accepted binomial and a synonym, or has multiple possible infraspecific matches. In these cases, the resulting matches will have a corresponding flag pointing the user to both the database/s and the type of conflict that was found.

Specifically, the highlight_flags() function returns three different flags for each backbone: infraspecific_ambiguity, authorship_ambiguity, and infraspecific_link. The infraspecific_link flag indicates that the input name was successfully resolved to a single Latin binomial, but at some point in the chain this involved linking a trinomial to a corresponding Latin binomial, which may or may not be appropriate depending on the taxon. For example, using WFO as the focal backbone, Abies shastensis is resolved at the species level to Abies magnifica, but it also returns an infraspecific_link flag because this result was linked through Abies magnifica var. shastensis. The infraspecifc_ambiguity flag indicates that the input binomial has a corresponding trinomial that would resolve to a different Latin binomial. For example, in WFO, Abies balsamea would raise this flag because Abies balsamea var. fraseri is a synonym of the accepted species Abies fraseri. Ignoring or deleting infraspecific epithets can thus create substantial name-resolution errors, and the user is advised never to naïvely truncate these beforehand. However, if the input data were incorrectly shortened from trinomial to binomial names at some previous point, this flag identifies instances where such shortening could lead to an incorrect binomial. Lastly, the authorship_ambiguity flag points out instances where a single input binomial corresponds to multiple binomial entries at taxonomic rank “species” in the related backbone. If resolving these entries would lead to different Latin binomials, the flag is raised, suggesting possible homonym issues at the species level. It’s important to note that the algorithm keeps the accepted name (e.g., Ilex subrotundifolia in WFO) and only selects the first entry in the backbone when neither of the homonyms is deemed accepted (e.g., Abies excelsa in WCVP), and is thus taxonomically arbitrary in that case. These entries should be treated with caution and should be manually verified.

Users are strongly encouraged to use this function after their matching and synonym resolving steps and determine whether they want to investigate these issues further. See ?highlight_flags for more details and examples.

Example Usage

Package Installation

library(devtools) install_github("speckerf/treemendous")

Species List Preparation

All functions of Treemendous require the species name to be split into two columns, Genus and Species, with the former being capitalized. Assume you have two species, Acer platanoides and Fagus sylvatica, you can create the input tibble by calling:

 ### Species list preparation library(tidyverse) species <- c('Acer platanoides', 'Fagus sylvatica') input <- species %>%   tibble::as_tibble_col(column_name = 'binomial') %>%   tidyr::separate(col = 'binomial', into = c('Genus', 'Species')) input

## # A tibble: 2 x 2 ##   Genus Species ##   <chr> <chr> ## 1 Acer  platanoides ## 2 Fagus sylvatica

Other useful functions for creating the input tibble include:

readr::read csv('path') # import data dplyr::select(Genus, Species) # select columns dplyr::distinct(Genus, Species) # remove duplicate binomials dplyr::rename('Genus' = 'old_genus_name',                 'Species' = 'old_species_name') # rename columns dplyr::mutate(Genus = stringr::str_to_title(Genus)) # capitalize Genus dplyr::mutate(Species = stringr::str_remove(Species, ".*?∖∖s")) # remove everything before first space tidyr::drop_na(c('Genus', 'Species')) # remove rows with NA's dplyr::arrange(Genus, Species) # sort names dplyr::bind_rows(x, y) # concatenate two tibble's

FIA: Standardize species names from the U.S. Forest Inventory and Analysis program.

Along with the package comes an example dataset fia with 2,171 different tree species names (Gray et al., 2012). Assume that we want to standardize these species names according to a certain backbone (use the backbone argument). The function summarize_output() can be used to get a summary of the process.

 library(treemendous)

 result <- fia %>% matching(backbone = 'BGCI') summarize_output(result)

## [1] "matched: 1822 / 2171 were matched with 1822 distinct matched names." ## [2] "direct_match: 1779 / 2171" ## [3] "indirectly matched: 43 / 392" ## [4] "    genus_match: 313 / 392" ## [5] "    fuzzy_match_genus: 2 / 79" ## [6] "    direct_match_species_within_genus: 1 / 315" ## [7] "    suffix_match_species_within_genus: 11 / 314" ## [8] "    fuzzy_match_species_within_genus: 31 / 303"

From 2,171 species names in total, we were able to match 1,822 according to the backbone BGCI, with 1,779 names matching exactly, and 43 species names matching using fuzzy- and suffix-matching. Besides information about the matching process, the output contains the old names (prefix Orig.) as well as the matched names (prefix Matched.) as follows:

 result %>%   dplyr::slice_head(n=3) %>%   dplyr::select(1:5)

## # A tibble: 3 x 5 ##   Orig.Genus Orig.Species Matched.Genus Matched.Species matched ##   <chr>      <chr>        <chr>         <chr>           <lgl> ## 1 Abies      amabilis     Abies         amabilis        TRUE ## 2 Abies      balsamea     Abies         balsamea        TRUE ## 3 Abies      bracteata    Abies         bracteata       TRUE

We can further increase the number of matched species by using the functions matching() followed by enforce_matching(). Here, we specify the backbone BGCI.

result <- fia %>%   matching(backbone = 'BGCI') %>%   enforce_matching(backbone = 'BGCI') result %>% summarize_output()

## [1] "matched: 2097 / 2171 were matched with 2036 distinct matched names." ## [2] "direct_match: 1779 / 2171" ## [3] "indirectly matched: 43 / 392" ## [4] "    genus_match: 93 / 117" ## [5] "    fuzzy_match_genus: 2 / 24" ## [6] "    direct_match_species_within_genus: 1 / 95" ## [7] "    suffix_match_species_within_genus: 11 / 94" ## [8] "    fuzzy_match_species_within_genus: 31 / 83" ## [9] "number of species matched via enforce_matching(): 275 / 349"

Now, we are able to match 2,097 species names in total, with 275 species being matched via enforce_matching(). Note that the number of matched distinct species names is lower with 2,036, because several input species were matched to the same species in the target database BGCI.

If we choose a different backbone than BGCI, then species can matched names that are not accepted (synonyms), we can further resolve synonyms after matching the species names with the function resolve_synonyms(). Now, the output contains additionally the accepted species names (prefix Accepted.), as well as a column Accepted.Backbone, which states according to which backbone the synonym was resolved.

 result <- fia %>%   matching('WFO') %>%   resolve_synonyms('WFO')

 result %>%   dplyr::slice_head(n=3) %>%   dplyr::select(dplyr::matches('Orig|Matched|Accepted'), -'matched')

## # A tibble: 3 x 7 ##   Orig.Genus Orig.Species Matched.Genus Matched.Species Ac~.Genus Ac~.Species ##   <chr>      <chr>        <chr>         <chr>           <chr>     <chr> ## 1 Abies      amabilis     Abies         amabilis        Abies     amibilis ## 2 Abies      balsamea     Abies         balsamea        Abies     balsamea ## 3 Abies      bracteata    Abies         bracteata       Abies     bracteata ## # i 1 more variables: Accepted.Backbone <chr> ## Abbreviated names: Ac~.Genus = Accepted.Genus, Ac~.Species = Accepted.Species

Note that a warning message is produced, “Please consider calling highlight_flags() to investigate potential ambiguities upon resolving synonyms to accepted names”. This indicates when potential ambiguities have been identified in your dataset, and it is suggested to use highlight_flags() to know more and decide if you want to check them manually. The highlight_flags() function should be used separately from the others as it will only return species that have some flag and not the full dataset. Here, we specify flags related to the WFO backbone:

 flags <- result %>% highlight flags('WFO')

## In summary, 574 out of 2171 matched species have raised a flag.

 flags %>%   dplyr::slice_head(n=3) %>%   dplyr::select(dplyr::matches('Acc|ambiguity|link'))

## # A tibble: 3 x 6 ##   Accepted.Genus Accepted.Species Accepted.Backbone WFO_authorship_ambiguity ##   <chr>          <chr>            <chr>             <lgl> ## 1 Abies          amabilis         WFO               TRUE ## 2 Abies          balsamea         WFO               FALSE ## 3 Abies          concolor         WFO               FALSE ## # i 2 more variables: WFO_infraspecific_ambiguity <lgl>, ## #   WFO_infraspecific_link <lgl>

We can see the full breakdown of these flags as follows:

flags %>% dplyr::select(dplyr::contains("WFO")) %>% dplyr::summarize all(.funs = sum)

## # A tibble: 1 x 3 ##   WFO_authorship_ambiguity WFO_infraspecific_ambiguity WFO_infraspecific_link ##                      <int>                       <int>                  <int> ## 1                      142                         462                     37

As we can see, the bulk of these flags denotes an infraspecific_ambiguity, which can generally be ignored, provided that the user did not manually truncate any trinomials to binomials for input, but should otherwise be investigated for those entries where this was done. The 37 infraspecific_link flags are likewise typically not problematic, as these simply highlight when the input binomial differs from the output binomial via a trinomial link at some point in the graph. The remaining 142 authorship_ambiguity are the most problematic, as these indicate taxa that have multiple conflicting matches. These should be manually explored and used with caution.

Instead of using a single backbone, the user can also decide to use any subset of the backbones c(‘BGCI’, ‘WFO’, ‘WCVP’, ‘GBIF’) or use all of them by simply calling matching() without any argument. While matching() considers all backbones being equally important, the function sequential_matching() can be used to call matching() for individual backbones sequentially. For every species, the matched backbone is provided in the column Matched.Backbone.

 result <- fia %>%   sequential_matching(sequential_backbones = c('BGCI', 'WFO', 'WCVP'))

Remember that matching() and sequential _matching() match any species in the database and thus can provide matches to synonyms rather than accepted species. To get only accepted species returned use resolve_synonyms() after the matching function.

Translate species names between two databases

Oftentimes, researches require integrating multi-modal data from different sources for their analyses. Here, we demonstrate the use of the function translate_trees(), which allows a user to directly translate names from an input database to a target database. First, we resolve both databases individually according to the single backbone(WFO) and compare the resolved names. Then, we use translate_trees() to translate the input species names into the target names.

 input <- tibble::tibble(   Genus = c('Aria', 'Ardisia', 'Malus'),   Species = c('umbellata', 'japonica', 'sylvestris') ) target <- tibble::tibble(   Genus = c('Sorbus', 'Ardisia', 'Malus'),   Species = c('umbellata', 'montana', 'orientalis') )

 input %>%   matching(backbone = 'WFO') %>%   resolve_synonyms('WFO') %>%   dplyr::select(1:6)

## # A tibble: 3 x 6 ##   Orig.Genus Orig.Species Matched.Genus Matched.Species Ac~.Genus Ac~.Species ##   <chr>      <chr>        <chr>         <chr>           <chr>     <chr> ## 1 Ardisia    japonica     Ardisia       japonica        Ardisia   japonica ## 2 Aria       umbellata    Aria          umbellata       Aria      umbellata ## 3 Malus      sylvestris   Malus         sylvestris      Malus     sylvestris ## Abbreviated names: Ac~.Genus = Accepted.Genus, Ac~.Species = Accepted.Species

 target %>%   matching(backbone = 'WFO') %>%   resolve_synonyms('WFO') %>%   dplyr::select(1:6)

## # A tibble: 3 x 6 ##   Orig.Genus Orig.Species Matched.Genus Matched.Species Ac~.Genus Ac~.Species ##   <chr>      <chr>        <chr>         <chr>           <chr>     <chr> ## 1 Ardisia    montana      Ardisia       montana         Ardisia   japonica ## 2 Malus      orientalis   Malus         <NA>            <NA>      <NA> ## 3 Sorbus     umbellata    Sorbus        umbellata       Sorbus    umbellata ## Abbreviated names: Ac~.Genus = Accepted.Genus, Ac~.Species = Accepted.Species

Resolving both sets individually leads to a mismatch—Malus orientalis and Malus sylvestris were resolved to two different names. Now let’s see whether translate_trees() can be used to match all three species:

 translate trees(df = input, target = target) %>%   dplyr::select(1:4)

## # A tibble: 3 x 4 ##   Orig.Genus Orig.Species Matched.Genus Matched.Species ##   <chr>      <chr>        <chr>         <chr> ## 1 Ardisia    japonica     Ardisia       montana ## 2 Aria       umbellata    Sorbus        umbellata ## 3 Malus      sylvestris   Malus         orientalis

Essentially, all three species names can be translated from the input set to the target set. Incorporating the knowledge of the desired target names, the function leverages the information about synonym-accepted relations in the three backbones WFO, WCVP and GBIF and is able to translate Malus sylvestris into Malus orientalis.

Discussion

The Treemendous package provides an efficient and reproducible approach to resolving species names and translating names between two disparate datasets. Although there are numerous taxonomic resolution packages, this approach is unique in that it leverages relationships across multiple backbones to increase the proportion of matched species consistent with a target backbone. In contrast to other applications, Treemendous adopts a graph-based approach that incorporates linkages and information across all backbones to further resolve species not present in a target backbone. By leveraging information across backbones, this package helps the user resolve or translate species lists that are inconsistent with any given backbone, improving matching success.

Treemendous is intended to provide a novel fourth step in the name-resolution pipeline, providing functionality not currently found in existing name-resolution packages. While this package also provides basic functionality for name cleaning and fuzzy-matching, it is intended to be used in tandem with—rather than in place of—existing packages. For example, pre-processing can also be done using packages and tools that specialize in one of these up-stream steps (e.g., Global Name Parser, TNRS, or Taxamatch), with resolve_synonyms() or translate_trees() subsequently applied to the parsed and cleaned Latin binomials after calling one of the matching functions. As such, Treemendous has the functionality to be used as a stand-alone package, but more ideally as the fourth step in resolving conflicting taxonomic lists.

A key challenge in taxonomic name resolution is the trade-off between false negatives (failing to successfully identify a match) and false positives (incorrectly identifying a match). The standard approach of most name-resolution applications is to tune this trade-off by toggling the number of allowable spelling mismatches when implementing fuzzy matching. Here, we provide an additional graph-based approach to this challenge, allowing the user to specify the search depth when implementing enforce_matching() (default of 3, Fig. 3), ranging from strict matching to minimize false positives (depth of 1) to more relaxed matching to decrease false negatives (depth of 3). Not only do these functions provide increased functionality and transparent control over the decision-making processes, but it allows the user to more easily find potential errors by identifying which species are sensitive to different resolution depths.

While linkages between backbones in the Treemendous were constructed using authority information and trinomials (variants, suspects, etc.), Treemendous currently requires a Latin binomial as input data. To identify potential issues, it returns a flag via highlight.flags() to indicate when an input Latin binomial has conflicting authority information or unclear Latin trinomial resolution. Of these flags, the infraspecific.link flag is generally benign, indicating that the accepted Latin binomial was reached via linking through a trinomial. The infraspecific.ambiguity flag is mainly of concern if the original dataset possibly or knowingly contained truncated trinomials. Because different subspecies and varieties with the same Latin binomial frequently resolve to different accepted Latin binomials, we reiterate that the user should never truncate trinomials, and instead should resolve these manually or with the aid of other packages. Last, the authorship.ambiguity flag highlights instances which resolve to different accepted Latin binomials, which often correspond to homonyms at the species level. In such instances, Treemendous returns the shortest path to an accepted binomial in the focal backbone, and otherwise simply picks the first match in the list, such that the selected species is taxonomically arbitrary. We strongly encourage the user to carefully explore the flagged taxa to determine the appropriate result, and to subsequently use another name-resolution application if needed, such as the recent U.Taxonstand package (Zhang & Qian, 2023), to incorporate authority information to help resolve these conflicts.

The core database used to resolve tree names is broadly inclusive (401,482 species names in total including synonyms and accepted names) and encompasses all species within a genus known to contain at least one characterized tree, which includes some woody shrubs and vines. Nonetheless, this package can also be used to identify and distinguish trees from other woody plants. Specifically, one can call matching() followed by enforce_matching(), with BGCI as the reference backbone. This will use the relations between synonyms in WFO, WCVP, and GFBI to translate the given species list into BGCI-consistent taxonomy (e.g., Fig. 3), ensuring that the resulting list contains only documented tree species. On the other hand, this package excludes non-woody tree-like species (e.g., tree ferns and monocots) which are absent from BGCI. For certain analyses (e.g., analysis of tree canopy) the inclusion of such species may be desirable, which presents an important next step for improving the extensibility of this package.

Treemendous is a fully open-source and collaborative tool that can easily be adapted to other taxonomic settings, including all plants or animals. Yet the challenge when moving beyond trees is the large number of possible species (e.g., ca. 375,000 known plant species (Christenhusz & Byng, 2016) vs. ca. 65,000 known tree species (Gatti et al., 2022)) which presents computational challenges, particularly when considering so-called ‘fuzzy matches’ (misspellings) across multiple backbones. To overcome some of the previous limitations, our approach uses fully vectorized functions, which avoids the need to process names sequentially; and it also relies on a built-in database, rather than requiring slow API calls by querying a remote database. Nevertheless, this package contains the core functionality for implementing taxonomic name resolution of other taxonomic groups, and would only require changing the reference database. We welcome and encourage any user suggestions or extensions to the package via the GitHub collaborative tools.

Taxonomic backbones are constantly being updated and revised, such that species lists that were previously consistent with a given backbone inevitably become outdated and inconsistent over time. By combining information across multiple backbones, the Treemendous package helps to overcome this challenge, allowing users to directly translate species names into a target species list, even when the target is inconsistent with any present-day backbone. By facilitating the integration of different data types and sources for biodiversity research, this functionality can help assimilate new data into existing workflows and models, increasing collaboration and data sharing across disciplines.

Supplemental Information

Supplemental Information 1 User manual

We thank Lalasia Bialic-Murphy for several discussions leading to the design of this package. We also thank Bradley L. Boyle and two anonymous reviewers for their insightful comments which substantially improved the manuscript and the package.

Additional Information and Declarations

Competing Interests

Author Contributions

Data Availability

The authors declare there are no competing interests.

Felix Specker performed the experiments, analyzed the data, prepared figures and/or tables, authored or reviewed drafts of the article, and approved the final draft.

Andrea Paz conceived and designed the experiments, authored or reviewed drafts of the article, and approved the final draft.

Thomas W. Crowther conceived and designed the experiments, authored or reviewed drafts of the article, and approved the final draft.

Daniel S. Maynard conceived and designed the experiments, authored or reviewed drafts of the article, and approved the final draft.

The following information was supplied regarding data availability:

The code and data are available at GitHub along with installation instructions and examples and Zenodo:

- https://github.com/speckerf/treemendous

- Felix Specker, Andrea Paz, Thomas Crowther, & Daniel S. Maynard. (2023). speckerf/treemendous: Zenodo Release v1.1.1 (zenodo-v1.1.1). Zenodo. https://doi.org/10.5281/zenodo.8251851

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
