# Peer review of "Treemendous: an R package for integrating taxonomic information across backbones"

_PeerJ, doi:10.7717/peerj.16896_

## Round 0.1 · original submission · Major Revisions

The three reviewers each identify key issues that need to be addressed. While all consider the underlying paper worthy of consideration, all have picked out errors and missing details that must be addressed. I would urge you to pay particular attention to identifying the limits of such automated resolution and the inability to process taxon concepts.

Reviewer 1 ·

Basic reporting

The overall writing is clear, and the structure is okay.
But the authors didn't include some recent developments in this field, such as these two recent papers:
(1) https://nph.onlinelibrary.wiley.com/doi/10.1111/nph.18961
(2) https://doi.org/10.1016/j.pld.2022.09.001

Experimental design

The method and the research question is clear for me, although I don't think that the question is important. Because that the other works have done much better jobs than this one.

Validity of the findings

no comment

Reviewer 2 ·

Basic reporting

The text is generally in clear unambiguous English, with good diagrams, literature, etc.

The only points that need adjusting are:

1. The codebox on line 258 indicates the use of the BGCI backbone. Subsequent text appears to be describing this codebox, but in line 260, the WFO backbone is mentioned. It is unclear if this is intended, or a typo.
2. Lines 120-128, there are several square brackets that appear not to have been closed.

The following should be considered suggestions only.

3. It is unclear what approach is being followed when backbones are updated. This is not detailed in the main text, or the GitHub readme. Reproducibility is mentioned in the text; being able to reproduce the exact version of a backbone used may be valuable to future users.
4. The diagrams could benefit from being re-drawn.
o It may be more appropriate to re-draw them following UML diagramming approaches.
o As presented, the text in the diagrams is very small, a re-arrangement with a larger font size within the diagram would aid accessibility.
5. The English in section 4 is a little less clear than the rest of the paper. As examples, “a certain genus” on line 185, and “process information” on line 200 are unclear.
6. A brief introduction should be added to the GitHub readme to help aid understanding. This would only need to be 2-3 sentences.

Experimental design

no comment

Validity of the findings

The points the authors are aiming to convey with the discussion are mostly clear. The main issue is that there could be a clearer and more explicit discussion about the benefits and potential issues (if there are any) with this package. For example, at the start of the start of the discussion, there is mention of the risk of false positives and negatives with backbone matching tools; the discussion would benefit from expanding on this point in more detail, illustrating how this tool relates to these issues.

Additional comments

In general, the paper is good, and the tool well designed. The logic of the tool is sensible, and should provide good value and benefit to its users. I am also pleased to see a nicely organised GitHub repo, with the readme only needing some minor additions to be very good. The main area that could be improved is the start of the discussion, where the authors should consider going into a little more explicit detail about the benefits and limitations of their package.

·

Basic reporting

In general this article is well written, in clear concise English, with only a few minor mistakes, as noted below. There are however two unfortunate errors in the Abstract which should be corrected: WCVP stands for "World Checklist of Vascular Plants", not "World Consensus on Vascular Plants". BCGI stands for "Botanic Gardens Conservation International", not "Botanical Gardens Convention International". These acronyms are later translated correctly (BCGI on line 136 and WCVP on line 153). Please update in Abstract.

Figures and tables are appropriate, and code and data are publicly available. However, I was unable to get the application to install on my local machine due to the large number dependencies and updates required (R is fragile; applying many package updates at once can break the installation. and require extensive troubleshooting to fix; I can't risk the downtime right now). I also attempted to install it on two of the linux servers I run, but some key dependencies were declared un-installable and the installation aborted; I can't risk breaking R for other users even temporarily). As far as I could tell without running the application, the code appears to be clean and efficient, making extensive use of Wickham's tidyverse tools and "missing" R features such as the magrittr pipe.

Literature citations are generally appropriate, but omit some important previous work. In particular, a key generalization—"Although there are numerous taxonomic name resolution packages, these rely on harmonizing species lists relative to a single reference backbone"—is unsupported by literature citations and readily rejected by inspection of the Global Names Resolver (https://verifier.globalnames.org/) and the Taxonomic Name Resolution Service (https://tnrs.biendata.org/; redirect from published url http://tnrs.iplantcollaborative.org). Note that the TNRS publication describes in detail the default approach for selecting a single "best match" from multiple candidate matches in multiple sources). See Boyle et al, 2013. BMC Bioinformatics 14.

Experimental design

The authors correctly identify gaps in single taxonomic sources and conflicts of taxonomic opinion among multiple sources as major remaining challenges for taxonomic names resolution. Their proposed solution, as implemented in the application Treemendous.Trees (hereafter, TT) is based on searching for connections among sources, is novel and represents an important advance in the field.

However, the impact of this novel approach could be increased by addressing the following shortcomings: (1) the application ignores name authors, thus discarding essential information for detecting rare but important sources of error, such as homonyms, (2) the authors need to highlight more clearly that the key difference between their application and other applications is *not* that TT resolves a single best match from among multiple sources (other applications do this; see comment above) but that it choses a single best match based on linkages among sources, (3) the authors fail to be candid that many important features of earlier applications are absent from TT, (4) stronger caveats are needed to caution the users agains accepting resolution results uncritically, and (5) the authors re-invent some major name resolution algorithms without recognition of existing approaches or explanation of how their approach is an improvement over earlier versions.

Regarding authors, all major taxonomic reference databases include authors, and most widely-used name resolution applications resolve authors—and for good reason. Identical names with different authors, when not merely minor spelling variants of the same author, are homonyms which point to different, often very different, taxa. In my lab, much of our current focus is on species distribution modeling using georeferenced species occurrence data. Occurrence records bearing illegitimate posterior homonym names, while rare, are highly consequential for species distribution modeling. Undetected occurrences of different, often phylogenetically distant, species can grossly distort modeling results.

It is essential that the authors make clear to readers that TT cannot be relied up to resolve correctly homonyms such as Ilex subrotundifolia C.J. Qi & Q.Z. Lin. The latter name, a posterior homonym of Ilex subrotundifolia Steyerm. (endemic to Venezuela), appears occasionally in Chinese forestry data. Its accepted name is Ilex elmerrilliana S.Y. Hu. I didn't test other applications, but the GN verifier and TNRS resolve this name correctly.

Unless I am mistaken, resolution of subspecies, varieties. etc. is not supported. Infraspecific taxa are common in specimen data, and occasionally present in tropical forestry data, which tends to be highly specimen-based. Identifications to the variety/subspecies level, when provided, can be valuable for resolving some synonyms.

Lines 40-42 ("existing R packages query these lists against a reference backbone in order to “translate” them into a common set of names, typically by identifying homotypic synonyms, name variants, and misspellings, and converting them into an accepted Latin binomial.") raise issues relevant to both points 2 and 3. These lines conflate the three main steps of name resolution. This confusion occurs elsewhere in the manuscript as well. The impact of this article would be increased by making clearer the distinctions between these steps, highlighting the steps where the TT algorithms represent improvements. These steps are: (1) name parsing, (2) name matching and (3) name resolution (sensu stricto).

Step 1 is not performed by TT; the user is required to submit genus and specific epithet as separated, pre-parsed name components. Raw names data ("in the wild") commonly contain a mix of concatenatated names of different taxonomic ranks, authors, nomenclatural annotations, annotations of uncertainty, etc. Many existing name resolution tools perform this step (in particular, the Global Names Parser, also used as a plug-in by other name resolution applications; see Mozzherin, et al. 2017. “gnparser”: a powerful parser for scientific names based on Parsing Expression Grammar. BMC bioinformatics, 18, pp.1-14.) TT cannot resolve such names. The authors should state clearly that resolving names data "in the wild" will require extensive ad hoc parsing by the user.

In step 2, the submitted string is matched to a documented scientific name present in one or more taxonomic references. This step can involve both exact and fuzzy matching, both of which are performed by TT. In step 3 the matched name is flagged as accepted, synonym or unusable, and, if a synonym, transformed to an accepted name. Examples of unusable names are rejected names (nom. rej.), invalid names and illegitimate names such as posterior homonyms (although the latter are occasionally linked to a plausible "intended" accepted name). In most name resolution applications, step 3 is a simple database lookup of the taxonomic status of the matched name in a single taxonomic source, followed by translation of synonyms to accepted names. "Identifying homotypic synonyms, name variants, […] and converting them into an accepted Latin binomial" belongs here.

The key point to highlight is that TT adds an important new "step 4" to the name resolution process: the interpretation of conflicting taxonomic opinions among multiple sources though the examination of graph relationships among multiple names in multiple sources. This is a major advance in the field, and should not be obscured by confusion with name parsing, name matching, or trivial database lookups of accepted names.

That said, the authors need to state more explicitly the limitations of their algorithms, and issue stronger cautions against their uncritical use. As the authors explain in the introduction, differences of taxonomic opinion among multiple reference taxonomies can lead to the persistence of multiple synonymous names for the same species in name resolution results. The solution implemented in TT is a graph-based approach which uncovers cross-linkages among names in different sources that may not be evident when only a single source is used. In particular, function enforce_matching() uncovers opposing linkages (e.g., source A say name 1 is a synonym of name 2, but source B says name 1 is a synonym of name 3) and circular linkages (e.g., source 1 says name 1 is a synonym of name 2, but source B says name 2 is a synonym of name 1). In addition, taxonomic sources can be ordered, such that a conflicting opinion in a lower-ranked source can be rejected in favor of the higher ranked source. The ability to discover and flag such conflicts is a novel and useful feature. However, the user must understand that the resulting selection of one opinion is over others is taxonomically and nomenclaturally arbitrary.

Function enforce_matching() goes beyond simply flagging conflicting opinions; it merges them, favoring one taxonomic opinion over another according to a decision algorithm that was not clear to me even after several readings. In the example shown in Fig. 3, submitted names Parinari racemosa, Atuna racemosa and Parinari laurina are all resolved to Maranthes corymbosa. Yet one source (WFO) treats the first and last of these names as synonyms of accepted species A. racemosa, and furthermore, is very clear in treating M. corymbosa as a separate accepted species. Yet, the WFO taxonomic decision to combine P. laurina and P. racemosa under accepted name Atuna racemosa is discarded simply because P. racemosa links WFO to WCVP, which treats the P. racemosa as a synonym of Maranthes corymbosa. It is not clear by what basis the WCVP decision to combine P. racemosa with Maranthes corymbosa trumps the WFO decision to keep them separate—other than, once that decision is made, Atuna racemosa is automatically lumped with Maranthes corymbosa as a homotypic synonym of Parinari racemosa. What is clear is that there is no basis in either the rules of nomenclature or taxonomic practice for this decision, without consulting addition information in the literature and not present in the TT database. Although the authors state that "the user is encouraged to manually check that these matches are also reasonable for their individual use case", they should be emphasize more strongly that these algorithmic decisions have no basis in nomenclature or biology and are as likely to be incorrect as correct. In my opinion, the user should also have the option of returning all matches with conflicting opinions flagged, as an aid to researching a preferred solution.

Finally, returning to step 2 (name matching), in their approach to fuzzy matching, the authors fail to acknowledge previous work on fuzzy matching of taxonomic names, in particular the sophisticated approach of Taxamatch documented in detail in Rees, 2014 (Taxamatch, an Algorithm for Near ('Fuzzy’) Matching of Scientific Names in Taxonomic Databases. PLoS One 9:e107510. https://doi.org/10.1371/journal.pone.0107510). The method is so effective that it has been used essentially as a "plugin" in other applications such as the Global Names Verifier, the TNRS and the WorldFlora R package (the latter two are both cited in this manuscript). While I highly encourage the development of new algorithms and approaches, doing so in ignorance of what has been done before can just as easily lead to worse rather than better outcomes. The authors should explain how their approach differs from Taxamatch, and demonstrate that it provide superior outcomes or performance. Above all, we need to respect the plea of Grenier et al (2021) that we avoid "re-inventing the wheel".

Validity of the findings

I am conflicted about this manuscript. One the one hand, the ability to detect and resolve conflicts of taxonomic opinion among different taxonomic references adds an important new step to the taxonomic name resolution workflow. One the other hand, the lack of major essential features long provided by other name resolution applications (such as name parsing and the ability to interpret and resolve homonyms) and the de novo replacement of existing solutions without justification (e.g., fuzzy matching) in my mind represents a step backward. It is also part of a troubling recent proliferation of name resolution applications in the R language which prioritize algorithm development at the expense of taxonomic and nomenclatural knowledge (Kindt 2020, WorldFlora is a notable exception). In many cases, such applications make it easier to get to a wrong result without knowing why or even noticing.

I think the present application sits somewhere in the middle better the latter and Kindt. Overall, I would like to see it publishes, assuming the shortcomings of the application are either remedied by updates to the code, or frankly acknowledged and accompanied by stronger caveats to avoid misuse of some features.

Additional comments

Lines 56-58. "This approach certainly increases the success rate, but it is ignorant of relationships among synonyms across backbones, leading to the potential presence of duplicate species and inconsistent species lists." This is true, and I commend the authors for tackling this important shortcoming of other applications.

Line 105. Change "tibble’s" to "tibbles"

Just a suggestion. Treemendous.Trees is both a dataset (tree species of the world, according to source BCGI) and a general purpose taxonomic name resolution application. Conflation of the tool with the data may diminish the impact of the article by turning off potential users looking for a broadly applicable tool and who may not be interested in an application built around an artificial pruning of a subset of the Tree of Life. The authors should consider assigning a more general name to the application, and treating the description of the building of Treemendous.Trees as an example use case.

---

## Round 0.2 · Minor Revisions

One reviewer has provided a final set of minor revisions and some of these will help clarify and improve the manuscript. Beyond that, the paper is acceptable for publication so no further review will be needed.

Reviewer 1 ·

Basic reporting

I think my concerns have been satisfactorily dealt with (as well as the other reviewer′s). To me the ms has significantly improved now and deserves publication.

Experimental design

no comment

Validity of the findings

no comment

·

Basic reporting

All comments under Additional comments.

Experimental design

All comments under Additional comments.

Validity of the findings

All comments under Additional comments.

Additional comments

For this second review, I commend the authors on their efforts to address the issues I have raised. Overall, they have done an adequate job of providing caveats in their manuscript regarding the capabilities of their application, and warning flags in the application itself to alert the user to potential issues requiring additional research or the use of external applications.

There are still a number of errors in the manuscript regarding features and capabilities of other name resolution applications, and some misleading statements concerning the potential issues implied by the warning flags. I highlight these in my line-by-line comments below. Assuming these remaining issues are addressed adequately, the manuscript should be suitable for publication, in my view.

Line numbers (when used) refer to the Track Changes version

1. Abstract, line 2. Change "There are numerous taxonomic name resolution packages for R" to " There are numerous taxonomic name resolution applications"
• Many major name resolution applications are not written in R, and many R packages are merely wrappers that query APIs of non-R applications.

2. Line 48. Change "cleaned" to "parsed"

3. Line 57. Change "R packages" to "applications"
• TNRS is not an R package. It is written in PHP, MySQL and Perl (https://github.com/ojalaquellueva/TNRSbatch; see also https://github.com/ojalaquellueva/tnrs_db). The TNRS R package (https://github.com/EnquistLab/RTNRS) is merely a convenience wrapper for R users, added years later, that queries the TNRS API (https://github.com/ojalaquellueva/TNRSapi). No major TNRS functionality is coded in R.
• The original TNRS described in the 2013 publication was written in a variety of other languages (none of them R), but the current version has been refactored in the languages listed above.

4. Lines 57-60. This is incorrect. Please re-word.
• The TNRS does all three steps, although step 1 is handled by Biodiversity (an early version of the GNParser) and parts of step 2 are handled by a PHP port of Taxamatch. I'd estimate <10% of the code is devoted to step 3, which is mostly a simple database lookup in SQL. From the user's perspective, the TNRS does all three steps.
• The integrated external applications responsible for steps 1 and 2 (the Biodiversity and TaxaMatch) are not "basic functionality" but are comprehensive solutions, deeply informed by taxonomic knowledge and expertise. I am not blowing my own horn here; I am referring to Dmitry Mozzherin, Tony Rees and co-developers such as David Shorthouse (a practicing systematist, in addition to expert coder).

5. Line 86. Change "R packages" to "Applications"

6. Line 86. I recommend citing one of the many earlier applications that do this. For example, Scott Chamberlain's taxize R package was developed for the purpose of resolving among multiple taxonomic backbones (currently, it consults something like 20; see https://ropensci.org/blog/2018/05/23/taxize-seven-years/).

7. Line 86. To be clear, the TNRS does not do this iteratively, it queries all sources simultaneously. This is evident if you use the web interface (tnrs.biendata.org) although the R wrapper does the same thing.

8. Lines 335-336. "This flag is therefore only of note if the user manually truncated any of the input trinomials to binomials and indicates where such truncating may be incorrect, but it otherwise can be ignored". This is misleading. In the absence of information about when or where the tree was observed or collected, you have no way of knowing if the binomial "Abies balsamea" refers to Abies balsamea sensu str. (found mostly in Canada and extreme NE USA) or Abies fraseri (endemic to the Appalachians). Numerous specimens of A. fraseri were labeled or re-labeled as Abies balsamea after 1982, when Murray published the new combination Abies balsamea subsp. fraseri (Pursh) A.E. Murray. Although A. fraseri was later resurrected (I believe by FNA) some specimens still appear in collections under Abies balsamea subsp. fraseri. It is entirely possible for Abies balsamea subsp. fraseri to get truncated to Abies balsamea, as many researchers routinely ignore infraspecfic epithets. Potential issues involving trinomial are not trivial and should never be ignored. Please correct.

9. Lines 337-339. "The authorship_ambiguity flag indicates a true problem with name resolution, highlighting input taxa that have multiple different matches". Please be more precise. If the authorship_ambiguity flag means that the binomial is associated with different authors, indicating potential homonym issues, you should say so. Also note that different authorship strings are not always a problem; in most cases, the differences are due to variations in punctuation, abbreviation or other conventions. For example, Elodea granatensis Bonpl, Elodea granatensis Humb. & Bonpl. and Elodea granatensis Humboldt & Bonpland are all the same thing.

10. Line 434. Change "packages" to "applications".

11. Lines 462-464. "Of these flags, infraspecifc_ambiguity is only of concern if any of the input binomials were obtained by manually truncating a trinomial, and can otherwise be ignored." Not true. Delete or correct. See comment 8.

12. Lines 465-467. "The authorship_ambiguity flag, however, is the most problematic flag, highlighting a fundamental conflict in name resolution within or across backbones." Not necessarily. Can be either a serious homonym issue or a trivial author spelling/abbreviation issue. See comment 9.

---

## Round 0.3 · accepted · Accept

I believe the last open comments and suggestions were all addressed and the manuscript is now in an acceptable format.